# Persistent Hearing Loss among World Trade Center Health Registry Residents, Passersby and Area Workers, 2006–2007

**DOI:** 10.3390/ijerph16203864

**Published:** 2019-10-12

**Authors:** James E. Cone, Cheryl R. Stein, David J. Lee, Gregory A. Flamme, Jennifer Brite

**Affiliations:** 1New York City Department of Health and Mental Hygiene, World Trade Center Health Registry, New York City, NY 10013, USA; jbrite@health.nyc.gov; 2School of Medicine, New York University, New York, NY 10016, USA; Cheryl.stein@nyumc.org; 3Department of Public Health Sciences, School of Medicine, University of Miami, Miami, FL 33199, USA; dlee@med.miami.edu; 4Stephenson & Stephenson Research and Consulting, Loveland, OH 45140, USA; gflamme@sasrac.com

**Keywords:** World Trade Center disaster, hearing loss, dust

## Abstract

Background: Prior studies have found that rescue and recovery workers exposed to the 9/11 World Trade Center (WTC) disaster have evidence of increased persistent hearing and other ear-related problems. The potential association between WTC disaster exposures and post-9/11 persistent self-reported hearing problems or loss among non-rescue and recovery survivors has not been well studied. Methods: We used responses to the World Trade Center Health Registry (Registry) enrollment survey (2003–2004) and first follow-up survey (2006–2007) to model the association between exposure to the dust cloud and persistent hearing loss (n = 22,741). Results: The prevalence of post-9/11 persistent hearing loss among survivors was 2.2%. The adjusted odds ratio (aOR) of hearing loss for those who were in the dust cloud and unable to hear was 3.0 (95% CI: 2.2, 4.0). Survivors with persistent sinus problems, headaches, PTSD and chronic disease histories had an increased prevalence of reported hearing problems compared to those without symptoms or chronic problems. Conclusions: In a longitudinal study, we observed an association between WTC-related exposures and post-9/11 self-reported hearing loss among disaster survivors.

## 1. Introduction

A previous publication from the World Trade Center Health Registry (Registry) regarding the cohort of individuals exposed to the events of the WTC disaster of 11 September 2001, found that dust cloud exposure among survivors who evacuated damaged and destroyed buildings was associated with self-reported hearing problems in a cross-sectional analysis [1]. A more recent publication from the Registry reported that among rescue and recovery workers, an increase in environmental hazards score and being unable to hear in the dust cloud were both independently associated with greater than double the odds of hearing problems [2]. A Fire Department of the City Of New York (FDNY) study found that firefighters and emergency medical service workers who were the most exposed to the WTC disaster had greater odds of persistent ear symptoms [3]. An analysis of medical surveillance audiometry data for FDNY firefighters and emergency medical service workers found that FDNY responders with high levels of exposure to the WTC rescue and recovery operations were at greater risk of reductions in hearing sensitivity persisting over the subsequent fifteen years, even after adjusting for aging [4]. 

The precise type of hearing loss and mechanism of persistent hearing loss associated with dust cloud exposure are not known. Extremely high noise exposure resulting from the building collapses and their aftermaths, ototoxic chemical exposure, infectious (sinusitis) and physical injury (concussion) all are plausible explanations for this association [5,6].

Reliable measurements of noise exposure during and following the disaster are mostly lacking. Potential ototoxic exposures that were detected in settled dust at the World Trade Center disaster site included heavy metals, polycyclic aromatic hydrocarbons, and other hydrocarbons [7]. Smoke and soot mainly contained byproducts of combustion of construction materials and building furnishings [7]. Other risk factors included exposure to welding and diesel exhaust fumes.

The purpose of this study was to conduct a longitudinal analysis to evaluate the association between exposure to the World Trade Center 11 September 2001 disaster and self-reported persistent hearing loss among residents, passersby, area workers, students and staff from schools in Lower Manhattan (survivors) who enrolled in the World Trade Center Health Registry in 2003–2004, and who responded to a follow-up survey in 2006–2007. 

## 2. Methods

### 2.1. Study Population

The study population has been described in detail elsewhere [8]. Briefly, the World Trade Center Health Registry was established as a prospective cohort study in 2002 and enrolled over 71,000 persons in 2003–2004. The Registry includes people present in Lower Manhattan on 9/11, rescue and recovery workers, lower Manhattan residents, passersby, area workers, children and students and staff from schools in the vicinity of the WTC site. 

For this analysis, we excluded enrollees who were rescue and recovery workers since they were examined separately, those interviewed by proxy and those with pre-9/11 hearing problems. The final analytic sample of survivors was 22,741 (Figure 1). Institutional Review Board approval was obtained from the New York City Department of Health and Mental Hygiene and the Centers for Disease Control and Prevention (CDC).

### 2.2. Exposure Assessment

Participants were asked in the Registry’s Wave 2 survey [9] whether or not they were in the dust cloud on 9 September 2001. If the participant answered ‘yes’, they were then asked a series of five yes-no questions, including, ‘When you were in the dust and debris cloud on 11 September 2001, which of the following did you experience? A. I could not see more than a couple of feet in front of me; B. I had trouble walking or finding my way because the dust was so thick; C. I had to find shelter like under a car or in a doorway; D. I was covered from head to toe with dust and debris; E. I could not hear anything.” The first (stem) question and this last question were used to create a three-tiered categorical measure of exposure, “Not in the dust cloud”, “In dust cloud, able to hear”, and “In dust cloud, not able to hear”. 

### 2.3. Outcome Assessment

The enrollment survey (Wave 1 [10] ) included the question, “Since 9/11 have you had a hearing problem?”. If the participant answered ‘yes’, this was followed by the question “Did you have a hearing problem before 9/11?”. If the participant reported a hearing problem prior to 9/11, they were excluded from further analyses. The Registry conducted the first follow-up survey of enrollees (Wave 2) in 2006–2007. In that survey, participants were asked to respond to the question, “Have you experienced a hearing problem or loss in the last 30 days?”. The outcome variable, persistent hearing problem was defined as those who indicated that they had new onset of a hearing problem following 9/11 on the Wave 1 survey (i.e., they had no pre-9/11 hearing problems) and also reported that they had experienced a hearing problem or loss in the previous 30 days on the Wave 2 survey.

### 2.4. Statistical Analysis

We performed a descriptive analysis of the proportion of survivors who had reported persistent hearing problems or loss by demographic, health status, and exposure factors. Comparisons were made between those with and without persistent hearing problems using standardized differences that estimate the differences between the means and proportions in each group divided by the standard error [11]. An imbalance in the covariate distribution between two groups is defined by an absolute value of the standardized difference being greater than or equal to 0.10, indicating at least a small effect size of a covariate [12]. 

We examined the association between WTC-related dust cloud exposure and hearing loss using logistic regression. We adjusted for mode (paper, phone or web at Wave 2), age at Wave 2 (integer), sex, race/ethnicity, education (less than high school, high school graduate, college graduate), smoking history at Wave 2 (never, former, current), persistent sinus symptoms at Wave 2 (yes, no), persistent headache symptoms at Wave 2 (yes, no), and chronic disease history (yes, no). Chronic disease history was a reported physician diagnosis of hypertension, heart disease, angina, heart attack or diabetes at Wave 1 or Wave 2. These conditions have been associated with hearing loss [13,14].

We ran several secondary analyses to evaluate the robustness of the observed associations. We restricted the sample to those 50 years or younger on 11 September 2001 to determine if the observed effects may be age-related. We did a stratified analysis for those with and without PTSD (defined as a Posttraumatic Stress Checklist—Civilian Version score of 44 or greater), and a separate stratified analysis by sex. We also used a more restrictive case definition based on responses to the Wave 2 question whether or not they had a medically diagnosed condition or disability that currently affected their hearing. 

Statistical analyses were performed using SAS 9.4 (SAS Institute, Cary, NC, USA).

## 3. Results

The prevalence of self-reported post-9/11 hearing loss or problem among survivors at the enrollment interview (2003–2004) was 1275/22,544 (5.7%). The overall prevalence of self-reported persistent hearing problems or loss (at both enrollment and follow-up surveys) among survivors in the Registry was 2.2%. Women reported a slightly higher proportion (2.4%) compared with men (2.0%) (Table 1). The mean age at Wave 2 of those with hearing problems was older (53.2 years) compared with those without (47.4 years) (*p* < 0.0001). 

The prevalence of hearing loss increased with higher levels of dust cloud exposure, from 1.3% among those not in the dust cloud to 2.2% for those in the dust cloud but able to hear, then to 5.7% for those in the dust cloud but unable to hear. Survivors with persistent sinus problems, headaches, PTSD and chronic disease histories had an increased prevalence of reported hearing problems compared to those without symptoms or chronic problems. 

In a fully adjusted model (Table 2), the adjusted odds ratio (aOR) for being in the dust cloud able to hear was 1.4 (95% CI: 1.1, 1.7). The aOR in the same model for being in the dust cloud unable to hear was 3.0 (95% CI: 2.2, 4.0). Asian survivors had a statistically increased aOR of 1.6 (95% CI: 1.1, 2.2) and Hispanic survivors had an increased aOR of 2.3 (95% CI: 1.7, 3.0) for persistent hearing loss.

In secondary analyses (data not shown), restricting the age at 9/11 of survivors to 50 years or less did not substantively change the overall pattern of the results. When the logistic regression analysis was stratified by PTSD at Wave 2, the magnitude of effect was reduced when limited to those with PTSD, and this was further reduced when limited to those without PTSD, although the dust cloud exposure measure remained statistically significant in both cases. 

When a separate case definition based on reported disability due to hearing loss was used, the overall pattern of results did not change substantively. 

## 4. Discussion

Similarly to our findings among 9/11 rescue and recovery workers, we observed an association between World Trade Center dust cloud exposure and reports of post 9/11 persistent hearing loss among World Trade Center disaster survivors, including Lower Manhattan residents, passersby, area workers, and school staff. In a fully adjusted model, those who could not hear while in the dust cloud on 9/11 had a three-fold increased odds ratio for persistent hearing loss compared to those not in the dust cloud. This relationship did not change significantly when secondary analyses were conducted limiting age to less than 50 years and stratifying by PTSD. 

The WTC dust, based on analyses of settled dust, was 80%–90% made up of a highly re-suspendable alkaline mixture of crushed concrete, gypsum (both known chemical irritants) and synthetic vitreous fibers, such as slag wool or fiberglass (known physical irritants) [15,16]. Much smaller fractions of the dust contained asbestos, trace metals, including lead and magnesium, organics such as combustion products, polycyclic aromatic hydrocarbons, polychlorinated biphenyls and dioxins [7]. Adverse health effects, including acute and chronic hearing loss, observed in persons exposed to WTC dust, were most likely due to the combined effects of high-level noise, the high alkalinity of the concrete and gypsum components and toxicity of the complex mixture of chemical constituents.

The reason why Asian and Hispanic survivors had a significantly increased adjusted odds ratio in this study is unclear. Prior studies of hearing loss after blast events rarely include information on race/ethnicity of the affected populations. A higher prevalence of hearing impairment has previously been found among US Spanish and Hispanic adults compared with the overall population, associated with noise exposure, socioeconomic factors and diabetes/pre-diabetes [17]. The prevalence of Asian American adults with hearing loss has been found to be comparable to other racial/ethnic groups [18]. 

PTSD without traumatic brain injury has been shown to be associated with hearing loss among Iraq and Afghanistan veterans [15]. The reason for this association is also unclear. Headache was a risk factor for persistent hearing loss in this study. Persistent hearing loss has previously been associated with reports of headache symptoms. Sudden sensorineural hearing loss has been associated with migraine [19]. 

High-level noise exposure was likely at the same time periods and locations where the dust cloud exposure occurred, although actual environmental measurements of both noise and airborne dust are lacking. Exposures with sound energy equivalent to 85 dBA for 8 hours exceed the recommended limits [20,21]. Excessive noise exposures may lead to permanent hearing loss, typically showing a ‘notch’ on the audiogram between 3000 and 6000 Hz. The cochlear portion of the inner ear is the most common site of permanent damage to the auditory system, but damage can also occur in the middle segments of the ear, the auditory nerve, or in the central nuclei of the auditory system [22]. Excess noise exposure primarily damages the delicate structures of the inner ear, particularly when the nutrient flow to the inner ear is compromised. The inner ear, auditory nerve, and central nuclei can be damaged by ototoxic chemicals. Genetic factors could possibly play a role as there is increasing evidence of the existence of single nucleotide polymorphisms predisposing to noise-induced hearing loss [23]. In addition, inflammation of nasopharyngeal tissues that often accompanies upper respiratory infections can lead to middle ear disorders. Although middle ear disorders tend to be temporary and can be treated with antibiotics, permanent hearing loss can be a consequence of repeated/chronic infections. Sinusitis, involving inflammation of nasopharyngeal tissues, is a common condition among WTC-exposed populations [24,25] and may also lead to conductive hearing loss [26]. Adverse otologic outcomes, including sensorineural or mixed hearing loss, tinnitus, vertigo tympanic perforation and hyperacusis, have been reported following previous blast injury incidents, including the Boston Marathon experience [5,27] and suicide bombings in Brussels [27]. 

The role of exposure to traumatic events such as 9/11 and their prolonged impact on physical and mental health outcomes is the subject of increasing focus by investigators [28,29,30,31]. The literature on potential mechanisms underlying associations between stress, psychological distress and hearing problems in humans is extremely limited [32,33,34]; there is also a need for more research in animal models examining the role of acute physical and psychological trauma and chronic stress on potential adverse effects on the cochlea and auditory cortex in order to identify mechanisms potentially amendable to intervention [35,36]. Finally, studies are needed to determine if administration of proven psychological and pharmacological post-trauma interventions and other stress management strategies might also serve to reduce adverse auditory system effects following future disasters [37,38].

Hearing loss is a major public health problem as it is the third most common chronic physical condition following hypertension and arthritis among United States adults [39]. The consequences of hearing loss may include depression [40], fatigue, social withdrawal [41], reduced physical activity [42], impaired memory, or headaches, which may result in an overall decreased quality of life and economic losses [43].

Strengths: This study is longitudinal and represents responses from a relatively large cohort of survivors. Simultaneous reporting of both exposure and outcome at two time points allowed a more detailed analysis of hearing loss among this disaster-exposed population of survivors than had been previously published.

Limitations: Loss to follow-up between surveys may have resulted in selection bias. Those who participated in Wave 2 tended to have fewer PTSD symptoms. This suggests that the prevalence of hearing loss at Wave 2 may be underestimated in this study. The self-reported nature of the survey data means that misclassification of exposure and outcome may have occurred. The type of hearing loss could not be determined based on the survey answers. We had no information about post-9/11 noise exposure from hobbies or other sources. 

The public health impact of hearing loss among survivors of the WTC disaster is an ongoing issue. The specific nature and causes of hearing loss among World Trade Center survivors remain to be determined. Analysis of pre- and post-9/11 audiograms among New York City Fire Department workers involved in health surveillance has been recently published [4]. However, similar surveillance pre- and post-disaster is missing for survivors. 

Hearing loss is not currently a certifiable WTC-related condition under the World Trade Center Health Program, making access to affordable treatment options potentially difficult. Health insurance typically does not cover hearing aids. 

## 5. Conclusions 

Public health officials should consider including questions about hearing trouble in surveys following future disasters involving chemical, dust, or blast exposures, and audiometric screening is recommended as part of ongoing surveillance for subsequent health effects.

## Figures and Tables

**Figure 1 ijerph-16-03864-f001:**
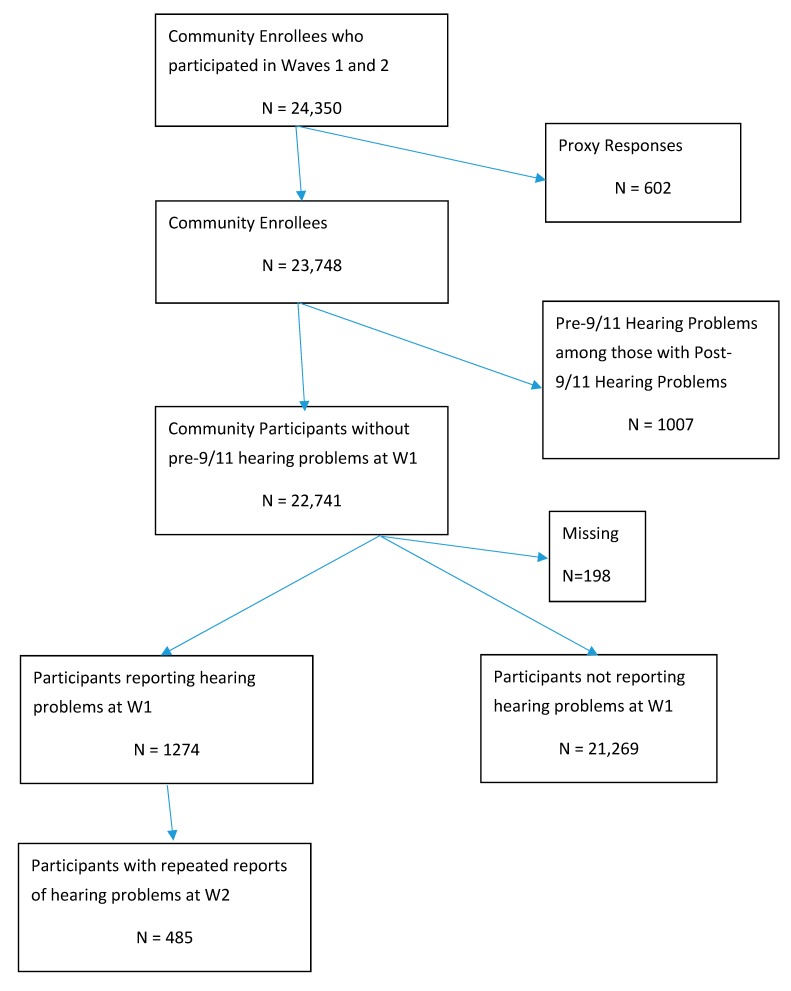
Flow chart of participation.

**Table 1 ijerph-16-03864-t001:** Characteristics of World Trade Center Health Registry community enrollees by persistent self-reported hearing problems ^a^, 2001–2007.

	Full Population (n = 22,741)			
	Persistent Hearing Loss			
Characteristic	Yes N (row percent)	No N (row percent)	Standardized Difference ^a^	Statistical Significance
Total (missing = 679)	485 (2.2)	21,577 (97.8)		
Dust Cloud Exposure			0.47	*p* < 0.0001
Not in dust cloud	116 (1.3)	8626 (98.6)		
In dust cloud, able to hear	242 (2.2)	10,870 (97.8)		
In dust cloud, not able to hear	102 (5.7)	1687 (94.3)		
Survey Mode, Wave 2			0.26	*p* < 0.0001
Web	167(1.6)	9989 (98.4)		
Phone	62 (2.2)	2751 (97.8)		
Paper	256 (2.8)	8837 (97.2)		
Age at Wave 2, years (mean, Standard Deviation)	53.2 (11.7)	47.4 (12.0)	0.49	*p* < 0.0001
Sex			0.08	*p* = 0.08
Female	280 (2.4)	11,602 (97.6)		
Male	205 (2.0)	9975 (98.0)		
Race/Ethnicity			0.44	*p* < 0.0001
Non-Hispanic White	227 (1.6)	14,135 (98.4)		
Non-Hispanic Black	72 (2.6)	2749 (97.5)		
Hispanic	122 (5.1)	2283 (94.9)		
Asian	46 (2.7)	1667 (97.3)		
Other	18 (2.4)	743 (97.6)		
Education			0.34	*p* < 0.0001
Less than high school	35 (6.2)	553 (93.8)		
High school graduate	207 (2.9)	6890 (97.1)		
College graduate	242 (1.7)	14,033 (98.3)		
Smoking History, Wave 2			0.06	*p* = 0.38
Never	259 (2.1)	12,154 (97.9)		
Former	153 (2.3)	6533 (97.7)		
Current	69 (2.5)	2725 (97.5)		
Sinus Symptoms ^b^			0.65	*p* < 0.0001
No	176 (1.2)	14,487 (98.8)		
Yes	300 (4.2)	6842 (95.8)		
Headache Symptoms ^b^			0.65	*p* < 0.0001
No	293 (1.5)	19,023 (98.5)		
Yes	178 (7.0)	2382 (93.1)		
PTSD Symptoms ^b^			0.75	
No	260 (1.4)	18,510 (98.6)		
Yes	170 (8.3)	1883 (91.7)		
Chronic Disease History ^c^			0.43	
No	191 (1.4)	13,126 (98.5)		
Yes	277 (3.3)	8132 (96.7)		

Note. SD = standard deviation, PTSD = post-traumatic stress disorder. ^a^ Standardized differences measure the effect size between two groups, independent of sample size [11]. ^b^ Sinus symptoms, headaches, and PTSD symptoms were reported at Wave 2. ^c^ Chronic disease history was a reported physician diagnosis of hypertension, heart disease, angina, heart attack or diabetes at Wave 1 or Wave 2.

**Table 2 ijerph-16-03864-t002:** Unadjusted and adjusted odds ratios (95% confidence intervals) for association between 9/11-related exposures and persistent self-reported hearing problems, World Trade Center Health Registry community enrollees, 2001–2007.

	Unadjusted OR (95% CI)	Adjusted OR (95% CI)
9/11 Exposure		
Not in dust cloud	1.0	1.0
In dust cloud, able to hear	1.7 (1.3, 2.1)	1.4 (1.1, 1.7)
In dust cloud, not able to hear	4.5 (3.4, 5.9)	3.0 (2.2, 4.0)
Mode Paper vs. web	1.7 (1.4, 2.1)	1.1 (0.9, 1.4)
Mode Phone vs. web	1.3 (1.0, 1.8)	1.2 (0.9, 1.7)
Age (years)	1.0 (1.0, 1.0)	1.04 (1.03,1.05)
Sex Male vs. Female	0.9 (0.7, 1.0)	1.1 (0.9, 1.4)
Race/ethnicity		
Non-Hispanic Black vs. Non-Hispanic White	1.6 (1.2, 2.1)	1.0 (0.73,1.4)
Asian vs. Non-Hispanic White	1.7 (1.2, 2.4)	1.6 (1.1, 2.2)
Hispanic vs. Non-Hispanic White	3.3 (2.7, 4.2)	2.3 (1.7, 3.0)
Other vs. Non-Hispanic White	1.5 (0.9, 2.5)	0.9 (0.5, 1.6)
Education < High School vs. College Graduate	3.8 (2.6, 5.5)	1.3 (0.8, 1.3)
Education High School Graduate vs. College Graduate	1.7 (1.4, 2.1)	1.0 (0.8, 1.3)
Smoking History-Current vs. Never	1.2 (0.9, 1.6)	1.0 (0.8, 1.3)
Smoking History-Former vs. Never	1.1 (0.9, 1.3)	1.0 (0.8, 1.3)
Sinusitis History	3.6 (3.0, 4.4)	2.3 (1.9, 2.9)
Headache History	4.9 (4.0, 5.9)	3.2 (2.5, 4.0)
Chronic Disease History	2.3 (1.9, 2.8)	1.3 (1.1, 1.6)

Note. OR = odds ratio; CI = confidence interval.

## Data Availability

Data are available upon reasonable request to the corresponding author.

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
