# Peer review of "Persistent Hearing Loss among World Trade Center Health Registry Residents, Passersby and Area Workers, 2006–2007"

_ijerph, 2019, doi:10.3390/ijerph16203864_

Round 1

Reviewer 1 Report

A typically expert presentation from this group.The subject matter is of great interest and not discussed as frequently as it should be.I have no revisions, and no reservations about publishing as is.

Author Response

Thank you for your review.

Reviewer 2 Report

Dear authors, your paper is well structured but some it is necessary to improve the discussion section, other than other structural changes in the material and methods. DISCUSSION You should discuss more (see my comment in the paper) as first the difference in risk between different ethnic groups, then due to to your main hypothesis (the effect of dust) you should discuss the dust contents and try to explain how each component may be responsible of the increased hearing loss. Maybe that some dust components act as toxic molecules... MATERIAL and METHOD Are these patients exposed to the noise during the attack? I think that would be helpful add details not only to the dust exposure, but also about the moment in which the people were exposed to dust. In case of subjects that were on WI place exactly during the impact moment, the noise exposure should be considered.

Author Response

Thank you for your comments and suggestions.

1. DISCUSSION You should discuss more as first the difference in risk between different ethnic groups.

We have added to the discussion the following to address the difference in risk for different ethnic groups:

"A higher prevalence of hearing impairment has previously been found among US Spanish and Hispanic adults compared with the overall population, associated with noise exposure, socioeconomic factors and diabetes/pre-diabetes [17]. The prevalence of Asian American adults with hearing loss has been found to be comparable to other racial/ethnic groups [18]."

2. Due to to your main hypothesis (the effect of dust) you should discuss the dust contents and try to explain how each component may be responsible of the increased hearing loss. Maybe that some dust components act as toxic molecules.

We have added a paragraph to address the components of the dust and their possible role:

"The WTC dust, based on analyses of settled dust, was 80-90% made up of highly resuspendable alkaline mixture of crushed concrete, gypsum (both known chemical irritants) and synthetic vitreous fibers such as slag wool or fiberglass (known physical irritants) [15] [16]. Much smaller fractions of the dust contained asbestos, trace metals including lead and magnesium, organics such as combustion products, polycyclic aromatic hydrocarbons, polychlorinated biphenyls and dioxins [7]. Adverse health effects, including acute and chronic hearing loss observed in persons exposed to WTC dust, were most likely due to the combined effects of high level noise, the high alkalinity of the concrete and gypsum components and toxicity of the complex mixture of chemical constituents."

MATERIAL and METHOD Are these patients exposed to the noise during the attack? I think that would be helpful add details not only to the dust exposure, but also about the moment in which the people were exposed to dust. In case of subjects that were on WI place exactly during the impact moment, the noise exposure should be considered.

We have added to the discussion the following sentence regarding high-level noise exposure:

"High-level noise exposure was likely at about the same time periods and locations where the dust cloud exposure occurred, although actual environmental measurements of both noise and airborne dust are lacking. "

We have made other changes to clarify the paper.

Thank you again for your comments and suggestions.